# GOAL: Grounded text-to-image Synthesis with Joint Layout Alignment Tuning

| Layout | Ours | | | GLIGEN | Attention Refocusing |
|---|---|---|---|---|---|

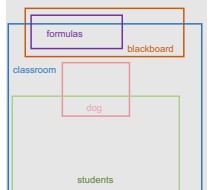 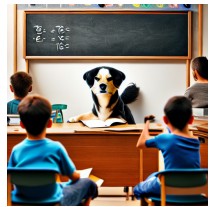 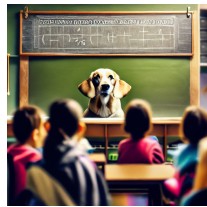 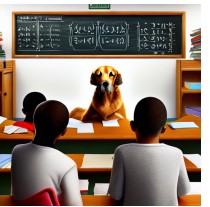 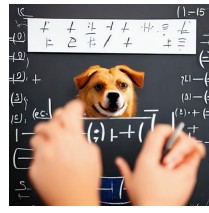 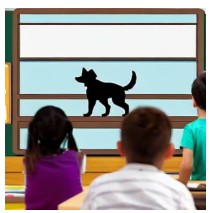

A dog is teaching math in a classroom, with the blackboard written with some mathematical formulas and the classroom packed with students.

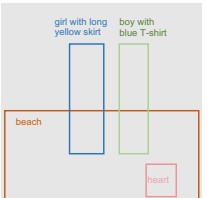 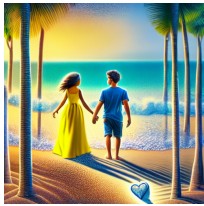 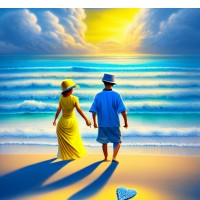 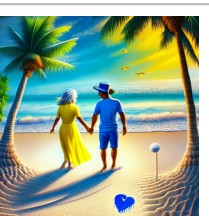 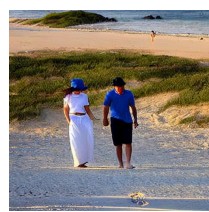 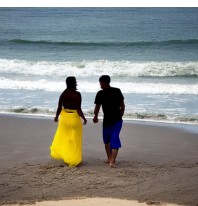

A girl, dressed in a yellow long skirt, is strolling along the beach with a man clad in a blue T-shirt. The sand is marked with a blue heart.

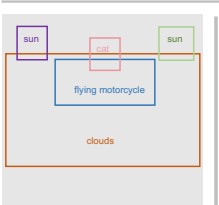 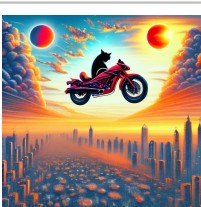 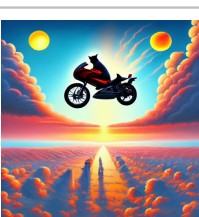 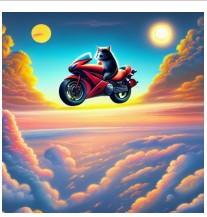 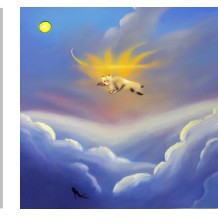 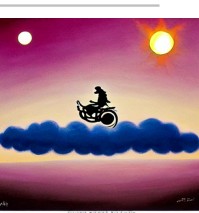

A painting of a cat riding a motorcycle flying over clouds, with two suns in the sky.

Figure 1: Given a user-specified text prompt and layout condition of complex scenes, baseline models (*e.g.*, GLIGEN [26], Attention Refocusing [34]) fail to solve text-image misalignment problems such as attribute errors and relation mistakes. Our method exhibits outstanding generation performance for text-to-image synthesis task, which generates images faithfully capturing the details of text prompts.

## ABSTRACT

Recent text-to-image (T2I) synthesis models have demonstrated intriguing abilities to produce high-quality images based on text prompts. However, current models still face Text-Image Misalignment problem (*e.g.*, attribute errors and relation mistakes) for compositional generation. Existing models attempted to condition T2I models on grounding inputs to improve controllability while ignoring the explicit supervision from the layout conditions. To tackle this issue, we propose Grounded jOint lAyout aLignment (GOAL), an effective framework for T2I synthesis. Two novel modules, discriminative semantic alignment (DSAlign) and masked attention alignment (MAAlign), are proposed and incorporated in this framework to improve the text-image alignment. DSAlign leverages discriminative tasks at the region-wise level to ensure low-level semantic alignment. MAAlign provides high-level attention alignment by guiding the model to focus on the target object. We also build a dataset GOAL2K for model fine-tuning, which composes 2000 semantically accurate image-text pairs and their layout annotations. Comprehensive evaluations on T2I-Compbench, NSR-1K, and Drawbench demonstrate the superior generation performance of our method. Especially, there are improvements of 19%, 13%, and 12% in color, shape, and texture metrics for T2I-Compbench. Additionally, Q-Align metrics demonstrate that our method can generate images of higher quality.

**Unpublished working draft. Not for distribution.**

*ACM MM, 2024, Melbourne, Australia*

## CCS CONCEPTS

• **Computing methodologies → Artificial intelligence**.

## KEYWORDS

Text-to-Image Generation; Diffusion Model; Large Language Model

## 1 INTRODUCTION

Recently, text-to-image (T2I) diffusion models have made tremendous progress in generating high-fidelity and diverse images in response to textual prompts [8, 13, 19, 40–42]. However, models still struggle with the Text-Image Misalignment problem [4, 9, 15, 29] for compositional T2I generation, which means they often fail to compose multiple objects with various attributes (*e.g.*, color, shape, texture) and complex spatial relations, as shown in the failure cases of GLIGEN [26] in Figure 1.

Previous works exploring compositional T2I generation can be classified into two main groups: training-free [9, 21, 34] and training-based approaches [26, 42, 52]. Specifically, training-free methods [9, 21] focus on directly altering the latent and cross-attention maps or adding objects by continuous editing [27, 53]. For instance, Attention Refocusing [9] employs inference-time optimization on the cross-attention map to align intermediate outputs with the layout conditions. However, it adds considerable cost during inference and may lead to image quality degradation, as shown in the last column of Figure 1.

Meanwhile, other training-based methods [42, 52] incorporate additional modules for controllable image generation. For example, as shown in Figure 2(b), GLIGEN [26] integrates grounding information into designed gated self-attention layers, supporting existing T2I models on layout inputs. Nevertheless, these layout-aware models ignore inherently explicit supervision in layout conditions, leaving them only as guidance for estimating the added global noise during training. We hypothesize that this objective is insufficient for complex generation tasks. Therefore, a natural question arises: Can we find fine-grained and explicit supervision with layout conditions as auxiliary training objectives to enhance text-image alignment? To achieve this, we adopt multiple phrase-region pairs in layout conditions as 'ground truth' labels, allowing the diffusion model to learn region-wise information within the image's context in an atomistic manner. Given that annotation of the "ground-truth" labels (*e.g.*, layout) from human is costly, we borrow large language model's (*e.g.*, GPT-4 [2]) strong scene understanding ability for layout planning.

In this work, we propose Grounded jOint lAyout aLignment (GOAL), an effective framework that utilizes layout conditions to provide explicit region-wise supervision for text-image alignment. It is achieved by incorporating discriminative semantic alignment (DSAlign) and masked attention alignment (MAAlign) as auxiliary training objectives. As shown in Figure 2(c), GOAL obtains a denoised version of the clean image via a single denoising step during training, DSAlign then utilizes discriminative tasks at the region-wise level to refocus on refining the generated context, ensuring low-level semantic alignment.

Considering that a more detailed denoised image can provide richer information for semantic alignment, we refine image details by applying region-wise MAAlign in high-level feature space. This

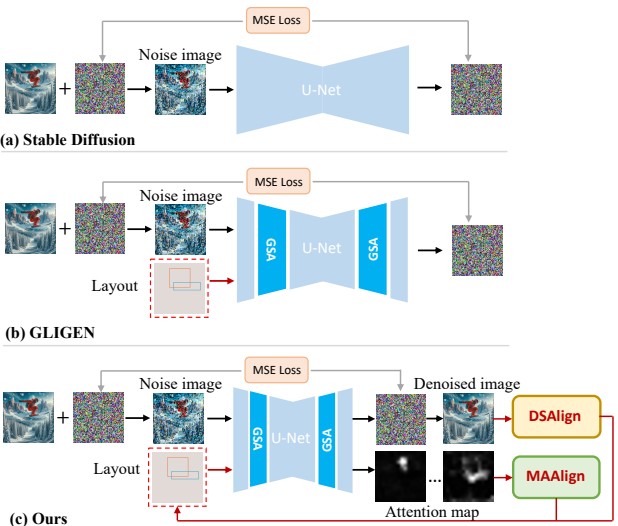

**Figure 2: (a) Stable Diffusion [40] is optimized to estimate the added noise with MSE loss. (b) GLIGEN [26] integrates grounding information into designed gated self-attention (GSA) layers, supporting existing T2I models on layout inputs. (c) GOAL provides explicit region-wise supervision by incorporating discriminative semantic alignment (DSAlign) and masked attention alignment (MAAlign).**

process involves shifting attention towards the target region while suppressing the attention of unrelated areas. As a result, sharper and more detailed images are generated, making semantic alignment more effective and further enhancing text-image alignment.

By conducting extensive experiments, the model exhibits significant improvement on several benchmarks, including T2I-Compbench [20], NSR-1K [16] and Drawbench [42]. Importantly, these enhancements are achieved without incurring additional inference costs. Along with our proposed training framework, we also publish a curated dataset GOAL2k for training, which consists of 2000 multimodal samples with layout annotation for improving text-image alignment especially in complex scenes. Moreover, images included in GOAL2k are generated by outstanding T2I model DALLE-3 [39], demonstrating both high-detail and semantically accurate characteristics.

The main contributions of this paper are summarized as follows:

- We propose a Grounded jOint lAyout aLignment (GOAL), which is a novel layout-aware training framework. Discriminative semantic alignment (DSAlign) and masked attention alignment (MAAlign) are incorporated in this framework to improve the text-image alignment.
- We build a dataset GOAL2K to study the effectiveness of our alignment-based objectives, which composes 2000 semantically accurate image-text pairs and their layout annotations for model fine-tuning.
- We conduct comprehensive experiments on existing methods of T2I generation on T2I-Compbench, NSR-1K and Drawbench, and show that our method compares favorably against the state-of-the-art models.

**Figure 3: An overview of the grounded joint layout alignment (GOAL) framework, which provides explicit region-wise alignment by discriminative semantic alignment (DSAlign) and masked attention alignment (MAAlign). GOAL obtains a denoised version of the clean image through a single denoising step and performs DSAlign by directly optimizing low-level semantic alignment. Furthermore, MAAlign is employed for high-level attention alignment, jointly optimizing the U-Net with DSAlign and its original denoising objective.**

## 2 RELATED WORKS

### 2.1 Text-to-Image Models

Early methods to text-to-image (T2I) generation primarily relied on Generative Adversarial Networks (GANs) [1, 12, 38, 44, 51]. However, recent advancements have shifted the focus towards diffusion models [8, 13, 19, 32, 45], which have gained prominence due to their exceptional capabilities in generating high-quality images. The Denoising Diffusion Probabilistic Model (DDPM) [19, 33, 46] introduces standard noise in the forward process and reconstructs the image from noise in the reverse process. Unlike denoising in the pixel space, the Latent Stable Diffusion Model (LDM) [40] conducts the denoising process in latent space, significantly reducing computational costs. Due to this remarkable progress, LDM has found wide application across various tasks, including image editing [7, 11, 18], image super-resolution [24, 43], inpainting [31, 50] and semantic segmentation [3, 6]. Building upon these state-of-the-art models, our work aims to enhance the capabilities of text-image alignment according to given layouts.

### 2.2 Training-Free Layout-Aware Generation

Despite the remarkable image generation capabilities demonstrated by Diffusion models, they encounter challenges in compositional generation, particularly within complex scenes. Recent methods [9, 34] primarily tackle this issue during the inference stage by incorporating layout-aware attention supervision at specific steps. BoxDiff [49] controls the noise map by adjusting cross-attention and self-attention layers. Attend-and-Excite [9] improves the generation of missing objects by maximizing the attention score for each object. Attention Refocusing [9] employs guidance functions to align intermediate outputs with layout conditions. Paint-with-words [5] enhances the cross-attention scores between image and text tokens corresponding to the same object based on segmentation masks. Continuous Layout Editing [53] disentangles various object concepts and facilitates continuous editing to align images with layouts. However, these methods incur computational costs during inference. Moreover, due to the direct optimization in the attention map without training, they may cause image distortion.

### 2.3 Training-Based Text-image Alignment

Several works aim to improve text-image alignment by fine-tuning diffusion models [10, 14, 26, 52] to integrate layout conditions. GLIGEN [26] integrates grounding information by the gated self-attention mechanism, enabling existing pre-trained T2I diffusion models to be conditioned on grounding inputs. LayoutLLM-T2I [36] introduces a relation-aware attention module, integrating semantic relation to generate high-fidelity images. Frido [14] performs multi-scale coarse-to-fine denoising to generate images of complex scenes. ReCo [52] incorporates spatial coordinates to achieve precise region control for arbitrary objects. Inspired by these works, we propose layout alignment at the semantic and attention level simultaneously to effectively fine-tune pre-trained T2I models without non-negligible costs.

## 3 METHOD

### 3.1 Preliminaries

Latent Diffusion Models (LDM) [40] are widely used in conditional image generative tasks. Given an image $x_0 \in \mathbb{R}^{H \times W \times 3}$, VAE $\mathcal{E}$ is adopted to encode image into latent space as $z_0 = \mathcal{E}(x_0)$. Then

Gaussian noise $\epsilon$ is added to the latent $z_0$ with a randomly sampled timestep $t$, yielding $z_t$ as:

$$z_t = \sqrt{\alpha_t}z_0 + \sqrt{1-\alpha_t}\epsilon, \tag{1}$$

where $\alpha_t$ defines the level of noise. To achieve conditional image generation, the caption describing the image is encoded by text encoder $\varphi$ as the text embedding $\varphi(y)$ and then injected into the U-Net via cross-attention. Then U-Net $\epsilon_\theta$ is trained to predict the added noise $\epsilon$, following the objective as:

$$\mathcal{L}_{LDM} = \mathbb{E}_{\mathcal{E}(x),y,\epsilon\sim\mathcal{N}(0,1),t}\left[\|\epsilon - \epsilon_\theta\left(z_t,t,\varphi(y)\right)\|_2^2\right]. \tag{2}$$

To ground the generation process via the additional condition, GLIGEN [26] incorporates the semantic information of grounding entity and spatial configurations through gated self-attention as:

$$v = v + tanh(\gamma) \cdot (SelfAttn[v,h]), \tag{3}$$

where $v$ is the visual feature, $\gamma$ is a learnable scalar which is initialized as 0 and $h$ is the added condition such as layout. Following GLIGEN [26], we optimize gated self-attention for stable training.

## 3.2 Discriminative Semantic Alignment

In the current layout-aware LDM, which incorporates additional layout conditions and optimizes the loss function $\mathcal{L}_{LDM}$ for noise prediction, the relationships within the bounding box, such as attributes and objects, are not explicitly optimized. As depicted in Figure 1, this results in poor semantic-level alignment in specific regions. Particularly, when different layouts overlap, the fusion of multiple conditions may result in a dissonant visual composition. Therefore, we propose discriminative semantic alignment (**DSAlign**), leveraging discriminative tasks at the region-wise level to refocus on refining the generated context. Specifically, with the noise predictor $\epsilon_\theta$ and $z_t$, we reconstruct the latent noise $z_0$ via a single denoising step using the reverse of Equation 1, and then obtain the denoised version of the clean image $x_0$ by VAE decoder $\mathcal{D}$. The formulation is:

$$\hat{x}_0^{(t)} = \mathcal{D}\left(\hat{z}_0^{(t)}\right) = \mathcal{D}(\frac{z_t - \sqrt{1-\alpha_t}\epsilon_\theta\left(z_t,y,t\right)}{\sqrt{\alpha_t}}). \tag{4}$$

Given the spatial layout defined by $B$ bounding boxes $b_i \in \left(\mathbb{Z}^+\right)^{1\times4}$, where $i \in [0,B)$, we perform DSAlign by minimizing the distance between the image embeddings of the target region and the corresponding phrase $p_i$ describing the region of $b_i$. To crop the region from the image, we construct the $M_i$ from the box $b_i$ and obtained the masked image as $\hat{x}_0^{(t)} \cdot M_i$, which is then encoded by the image encoder of CLIP, denoted by $CLIP_{img}$. We adopt the text encoder from CLIP, denoted by $CLIP_{text}$, to encode the corresponding phrase captions $p_i$. Then a normalization operation is performed to align the two embeddings within a unified semantic space. This process is formulated as follows:

$$\bar{x}_{0,m_i}^{(t)} = norm(CLIP_{img}(\hat{x}_0^{(t)} \cdot M_i)), \tag{5}$$

$$\bar{p}_i = norm(CLIP_{text}(p_i)). \tag{6}$$

Subsequently, the distance between the two normalized embeddings is computed using spherical distance as follows:

$$D_{sp}\left(\bar{p}_i, \bar{x}_{0,m_i}^{(t)}\right) = \arcsin\left(\frac{\|(\bar{p}_i) - (\bar{x}_{0,m_i}^{(t)})\|_2}{2}\right)^2. \tag{7}$$

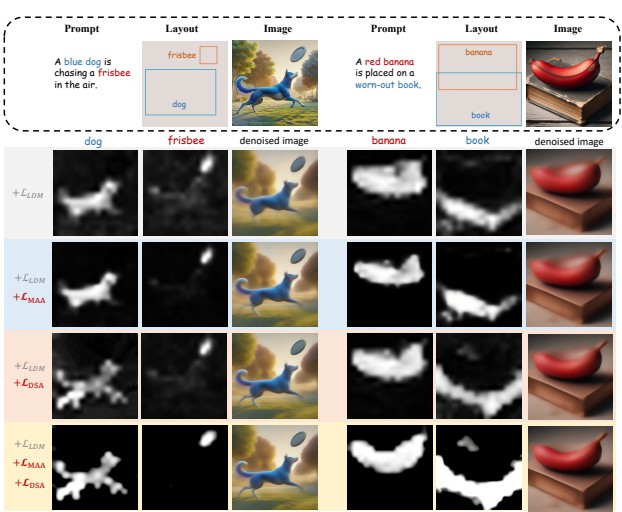

**Figure 4: Visualization of the attention maps and denoised images obtained from a single denoising step during training with different loss compositions.**

The average distance between all the $B$ target regions and the corresponding phrases is:

$$D_{local} = \frac{1}{B}\sum_0^B D_{sp}\left(\bar{p}_i, \bar{x}_{0,m_i}^{(t)}\right), \tag{8}$$

where $D_{local}$ represents the local-level semantic alignment guidance between target regions and local phrases. Moreover, we evaluate the global-level semantic alignment between the entire image and the text as follows:

$$D_{global} = D_{sp}\left(\bar{y}, \bar{x}_0^{(t)}\right) = \arcsin\left(\frac{\|(\bar{y}) - (\bar{x}_0^{(t)})\|_2}{2}\right)^2, \tag{9}$$

where $\bar{y} = norm(CLIP_{text}(y))$ and $\bar{x}_0^{(t)} = norm(CLIP_{img}(\hat{x}_0^{(t)}))$. To ensure the text-image alignment, we directly optimize local-level and global-level semantic alignment by $\mathcal{L}_{DSA}$:

$$\mathcal{L}_{DSA} = D_{local} + D_{global}. \tag{10}$$

## 3.3 Masked Attention Alignment

Given that a more detailed denoised image can provide richer information for semantic alignment performed by DSAlign, we refine image details through region-wise masked attention alignment (MAAlign) in high-level feature space. This process involves shifting attention towards the target region while suppressing the attention of unrelated areas. Specifically, for each cross-attention layer, let $Q \in \mathbb{R}^{hw\times d}$ be the input intermediate feature to the cross attention layer, which is obtained from the feature map of size $h \times w$ with feature dimension $d$, and let $K \in \mathbb{R}^{n\times d}$ be the transformed text embedding with $n$ tokens via a linear map, we can obtain cross-attention map $A^t$ at the step $t$ as follows:

$$A^t = \text{softmax}\left(\frac{\mathbf{Q}\mathbf{K}^\top}{\sqrt{d}}\right). \tag{11}$$

To ensure high responses within the masked regions, we shift attention towards the target regions by boosting the values of the masked attention map:

$$\mathcal{L}_{MAA-in} = \frac{1}{B} \sum_{i \in B} \left( 1 - \max \left( A_j^t \cdot M_i \right) \right), \quad (12)$$

where $j \in [0, n)$ is the token index describing the content of $M_i$, and $A_j^t$ represents the corresponding attention map. Moreover, we suppress attention on irrelevant areas by preventing attention from extending beyond the target regions:

$$\mathcal{L}_{MAA-out} = \frac{1}{B} \sum_{i \in B} \max \left( A_j^t \cdot (1 - M_i) \right). \quad (13)$$

The overall masked attention alignment loss is defined as:

$$\mathcal{L}_{MAA} = \mathcal{L}_{MAA-in} + \mathcal{L}_{MAA-out}. \quad (14)$$

In this way, MAAlign provides attention alignment by guiding the model to focus on the target object. Finally, the training objective is as follows:

$$\mathcal{L} = \mathcal{L}_{LDM} + \alpha \mathcal{L}_{DSA} + \beta \mathcal{L}_{MAA}, \quad (15)$$

where $\alpha$ and $\beta$ are trade-off parameters to balance the contributions of the $\mathcal{L}_{DSA}$ and $\mathcal{L}_{MAA}$.

Figure 4 illustrates the attention map $A^t$ and denoised image $\hat{x}_0^{(t)}$ with various combinations of loss terms. It's clear that when employing $\mathcal{L}_{DSA}$ (3rd row), the cross-attention maps exhibit a closer alignment with the real image. Furthermore, when adding $\mathcal{L}_{MAA}$ alongside $\mathcal{L}_{DSA}$ (4th row), the maximum value of the attention map is boosted, indicating an enhanced focus on the target object, thus resulting in a clearer denoised image. This enhances the effectiveness of DSAlign and further improves text-image alignment.

## 4 EXPERIMENT

### 4.1 Dataset Construction

To study the effectiveness of our alignment-based objectives, we construct a compact yet robust dataset called **GOAL2k** to fine-tune our model. Initially, we first select common objects from COCO dataset [28]. Subsequently, leveraging in-context learning, we devise various templates encompassing different categories (*e.g.*, color, shape and spatial). Then we employ GPT-4 [2] to generate relational captions according to the templates and objects, resulting in a set of 1,000 template-based prompts. Additionally, we directly choose 1,000 captions from COCO dataset [28] that contain multiple objects or spatial relations as natural prompts. To guarantee the high quality and semantic accuracy of images incorporated in GOAL2K, we employ DALLE-3 [39] for generating images in the training set. Subsequently, we utilize GroundingDINO [30] to produce layout annotations. Ultimately, we manually verify the generated image-text pairs alongside the layout annotations to ensure their fidelity to accurate semantic information. **More details about dataset construction can be found in supplemental materials.**

### 4.2 Evaluation Metrics

Building upon existing works [16, 20, 35], we evaluate the precision of layout-to-image generation across various widely-adopted benchmarks, such as T2I-CompBench [20], NSR-1K [16] and Drawbench [42]. For the evaluation of text-image alignment, we adopt the recommended protocols in T2I-CompBench [20]. For NSR-1K [16], we utilize cross-modal similarities, indicated by CLIP score [37], and detection-based similarities, denoted by GLIP score [25]. As Drawbench [42] doesn't provide labels or metrics for automatic assessment, we conduct evaluator-based evaluation via user study. To evaluate the image generation quality, we calculate Q-Align [48] as image quality score and aesthetic score. Additionally, PickScore [22] is employed to evaluate human preferences regarding the quality of generated images. Furthermore, NSR-1K [16] is also utilized to evaluate the accuracy of the layout generated by different text-to-layout models [2, 17, 47] across spatial and counting scenarios.

### 4.3 Implementation Details

Our experiments are performed using GLIGEN [26], a popular layout-to-image diffusion model built upon Stable Diffusion v1.4 [40]. GPT-4 [2] is employed for layout planning during inference. We use $\mathcal{L}_{DSA}$ and $\mathcal{L}_{MSA}$ as alignment-based objectives alongside the original denoising objective $\mathcal{L}_{LDM}$ to train our model. The weights ($\alpha$ and $\beta$) for these objectives are set as 1e-2 and 1e-3, respectively. CLIP ViT-L/14 [37] is employed in discriminative semantic alignment (DSAlign) as the encoder for the image and text. We select the attention map with size $16 \times 16$ obtained from the decoder of the U-Net to perform masked attention alignment (MAAlign). We employ the AdamW optimizer with a constant learning rate of 5e-5 over 14,000 steps, with a batch size of 1. During training, we fine-tune the gated self-attention layers following GLIGEN [26]. Additionally, to fully exploit DSAlign and MAAlign, we fine-tune both self-attention and cross-attention layers.

### 4.4 Quantitative results

To evaluate the text-image alignment, we compare the proposed method with other state-of-the-art methods in two aspects: with text-to-image (T2I) methods such as Stable Diffusion v1.4 [40], DPT [35], Attn-Exct v1 [9], and layout-to-image methods such as GLI-GEN [26], LayoutLLM-T2I [36], LayoutGPT [16] on T2I-Compbench [20]. Table 1 illustrates that our proposed method surpasses all other methods in all metrics. Particularly, GOAL achieves improvements of 19%, 13%, and 12% in color, shape, and texture metrics respectively, compared to GLIGEN. This verifies that discriminative tasks utilized by DSAlign could ensure region-wise semantic alignment between objects and corresponding attributes, and attention alignment performed by MAAlign allowing the model focusing on the target object, which is a prerequisite for attribute binding.

Additionally, we present the results of counting and spatial evaluation on the NSR-1K [16] benchmark in Table 2. The improvement, particularly in the GLIP [30] score (4% in counting and 3% in spatial metrics), further demonstrates the effectiveness of our method. From Table 1 and Table 2, we observe that layout-to-image methods outperform T2I methods in spatial and counting relationships by a large margin. This illustrates that employing layouts as an intermediate representation can significantly improve the controllable generation, particularly in scenarios involving counting and spatial relationships.

Moreover, we demonstrate the results of the proposed method compared with training-free methods such as Structure v2 [15], BoxDiff [49], Attend-and-Excite [9], and Attention Refocusing [34]

**Table 1: Comparison to the state-of-the-art text-to-image methods and layout-to-image methods on T2I-Compbench.**

| Method | Color | Shape | Texture | Spatial | Non-Spatial | Complex |
|--------|-------|-------|---------|---------|-------------|---------|
| *Text-to-Image Methods* | | | | | | |
| Stable v1.4 [40] | 37.65 | 35.76 | 41.56 | 12.46 | 30.79 | 28.18 |
| HN-DiffusionITM [23] | 36.71 | 35.48 | 39.84 | 11.22 | 30.91 | 28.05 |
| Composable v2 [29] | 40.63 | 32.99 | 36.45 | 8.01 | 29.80 | 28.98 |
| DPT [35] | 48.84 | 38.93 | 50.1 | 14.63 | 30.83 | 30.05 |
| DPT+SC [35] | 51.51 | 39.61 | 49.38 | 15.45 | 30.84 | 30.29 |
| *Layout-to-Image Methods* | | | | | | |
| GLIGEN [26] | 34.41 | 38.61 | 46.34 | 35.42 | 30.42 | 28.96 |
| LayoutGPT [16] | 33.86 | 36.35 | 44.07 | 35.06 | 30.31 | 26.36 |
| LayoutLLM-T2I [36] | 37.98 | 39.78 | 47.62 | 31.97 | 29.24 | 27.66 |
| Ours | **53.55** | **51.19** | **58.37** | **37.28** | **30.94** | **32.48** |

**Table 2: Comparison to the state-of-the-art text-to-image methods and layout-to-image methods on NSR-1K.**

| Method | Numerical Reasoning | | SpatialReasoning | |
|--------|---------------------|---------------|------------------|---------------|
| | Acc. (GLIP) | CLIP Sim. | Acc. (GLIP) | CLIP Sim. |
| *Text-to-Image Methods* | | | | |
| Stable v1.4 [40] | 32.22 | 0.256 | 16.89 | 0.252 |
| Stable v2.1 [40] | 42.44 | 0.256 | 17.81 | 0.256 |
| Attn-Exct v1 [9] | 38.96 | 0.258 | 24.38 | 0.263 |
| Attn-Exct v2 [9] | 45.74 | 0.254 | 26.86 | 0.264 |
| *Layout-to-Image Methods* | | | | |
| LayoutLLM-T2I [36] | 57.89 | 0.261 | 49.25 | 0.267 |
| LayoutGPT [16] | 55.64 | 0.261 | 60.64 | 0.268 |
| Attention Refocusing [34] | 57.26 | 0.244 | 61.23 | 0.251 |
| GLIGEN [26] | 56.02 | 0.258 | 60.03 | 0.265 |
| Ours | **60.25** | **0.263** | **63.12** | **0.271** |

**Table 3: Comparison to the state-of-the-art training-free methods on T2I-Compbench.**

| Component | Color | Shape | Texture | Spatial | Time (s) |
|-----------|-------|-------|---------|---------|----------|
| Structured v2 [15] | 49.90 | 42.18 | 49.00 | 13.86 | 4.23 |
| Box Diff [49] | 50.26 | 45.11 | 53.18 | 33.01 | 4.11 |
| Attn-Exct v1 [9] | 53.31 | 38.51 | 56.13 | 10.06 | 5.13 |
| Attention Refocusing [34] | 45.38 | 43.04 | 50.62 | 36.01 | 5.29 |
| Ours | **53.55** | **51.19** | **58.37** | **37.28** | **2.03** |

**Table 4: Comparison to the state-of-the-art text-to-layout methods on PickScore test-unique set.**

| Method | PickScore | Quality | Aesthetic |
|--------|-----------|---------|-----------|
| Stable v1.4 [40] | 0.2481 | 3.79 | 3.08 |
| Attn-Exct v1 [9] | 0.2152 | 3.71 | 2.89 |
| Attention Refocusing [25] | 0.1518 | 3.83 | 2.83 |
| GLIGEN [26] | 0.1686 | 3.96 | 3.05 |
| Ours | 0.2561 | 4.42 | 4.14 |

## 4.5 Qualitative results

Figure 5 demonstrates some cases from the T2I-Compbench [20] and NSR-1K [16] benchmarks across attributes such as color, spatial layout, shape, counting, and texture. We observe that 1) Layout conditions, serving as an intermediate representation, can enhance the generation controllability in scenes involving counting and spatial relationships. As shown in the last row, the T2I models Stable Diffusion [40] and Attend-and-Excite [9] fail to generate the correct number of animals, while other layout-to-image models achieve better control over the counting through guidance from the layout. 2) Our proposed method demonstrates outstanding performance across all attributes, generating images that faithfully capture semantic details. This can be attributed to the incorporation of DSAlign and MAAlign, which ensure region-wise semantic and attention alignment, respectively. 3) Compared with other methods, our proposed method can generate high-quality and aesthetic images, which verifies the improvement in text-image alignment does not result in a loss of image quality.

## 4.6 Ablation studies

**Effect of loss terms.** To explore the contribution of the loss terms $\mathcal{L}_{DSA}$ and $\mathcal{L}_{MAA}$, we conduct experiments in T2I-Compbench [20] including color, shape, texture, spatial metrics. From Table 5, we could see that both $\mathcal{L}_{DSA}$ and $\mathcal{L}_{MAA}$ could consistently promote alignment performance for T2I gengeration. Specifically, $\mathcal{L}_{DSA}$ shows considerable improvement in shape and texture metrics, while $\mathcal{L}_{MAA}$ exhibits significant enhancement in color metrics. It can be attributed to the fact that discriminative loss provided by semantic alignment improves shape and texture metrics, whereas attention-level alignment guides the model to focus on the target

on T2I-Compbench. Additionally, the inference time for generating an image is presented in Table 3. While these methods do not need additional training, they incur considerable computational costs during inference. For instance, Attention Refocusing [34] and Attend-Excite [9] take ×2.61 and ×2.52 times longer, respectively, to generate a single image compared to the proposed method. Our proposed method enables efficient inference and consistently outperforms baseline methods across all evaluation metrics.

Beyond the above evaluation, we also assessed the quality of images generated by different models using the PickScore [22] test-unique set. Results suggest that training-free methods, which directly intervene with cross-modal attention and latent noise maps during sampling, may degrade image quality. For instance, the PickScore of Attend-Excite [9] decreased by 3% compared to its baseline model Stable Diffusion v1.4 [40]. In contrast, our proposed method demonstrates exceptional performance for both text-image alignment and image quality.

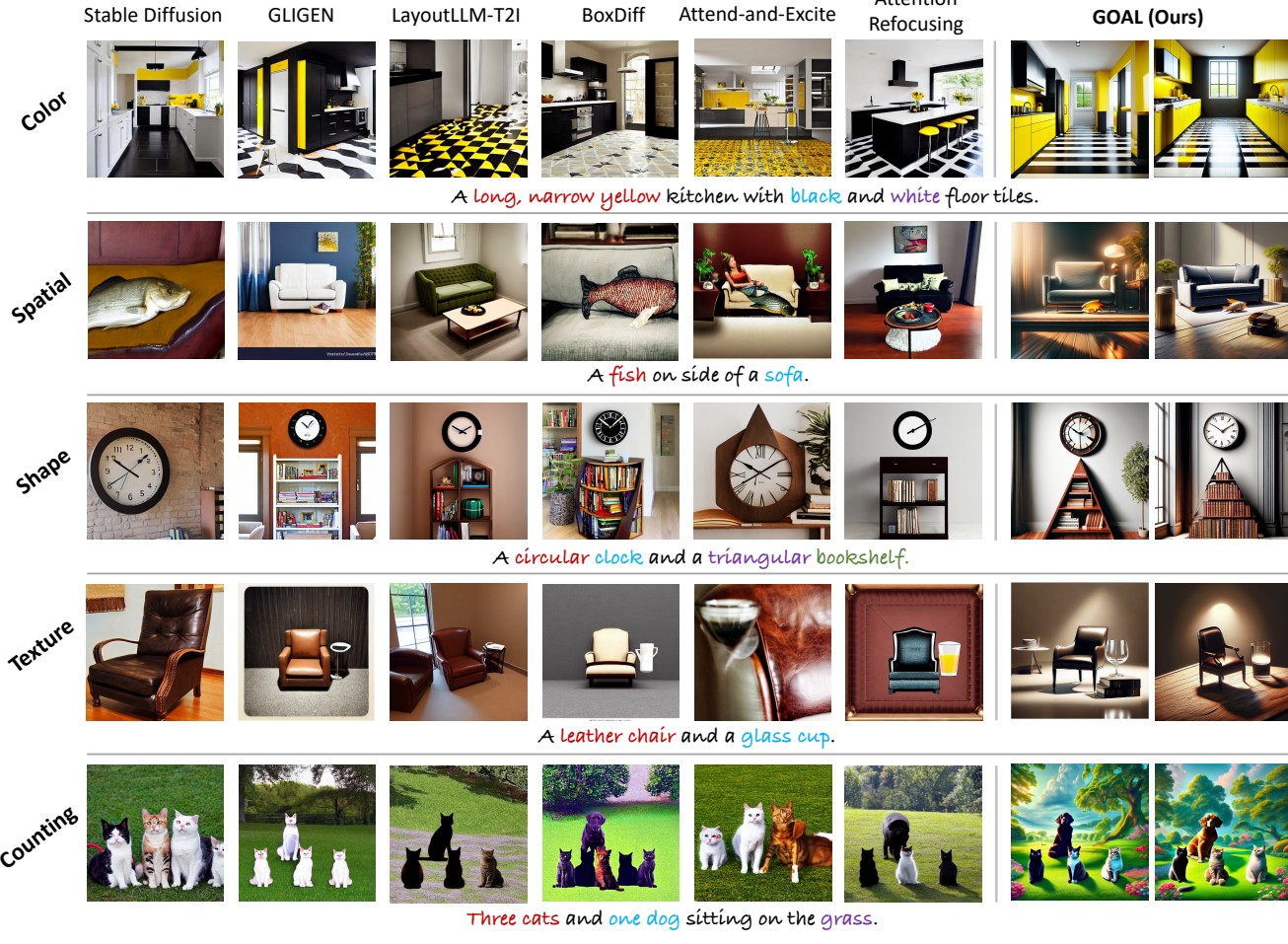

**Figure 5: Qualitative results from T2I-Compbench and NSR-1K for various attributes such as color, spatial, shape, counting and texture. We demonstrate the effectiveness of the proposed method in text-image alignment compared with Stable Diffusion v1.4 [40], GLIGEN [26], LayoutLLM-T2I [36], BoxDiff [49], Attend-and-Excite [9] and Attention Refocusing [34].**

**Table 5: Effect of loss terms.**

| Component | Color | Shape | Texture | Spatial |
|---|---|---|---|---|
| frozen | 34.41 | 38.61 | 46.34 | 35.42 |
| $\mathcal{L}_{LDM}$ | 44.17 | 44.27 | 53.86 | 36.42 |
| $\mathcal{L}_{LDM} + \mathcal{L}_{feature}$ | 49.83 | 46.89 | 55.21 | 36.77 |
| $\mathcal{L}_{LDM} + \mathcal{L}_{pixel}$ | 47.69 | 50.01 | 56.28 | 37.02 |
| $\mathcal{L}_{LDM} + \mathcal{L}_{pixel} + \mathcal{L}_{feature}$ | **53.55** | **51.19** | **58.37** | **37.28** |

**Table 6: Effect of different level of semantic alignment.**

| $D_{local}$ | $D_{global}$ | Color | Shape | Texture | Spatial |
|---|---|---|---|---|---|
| | | 49.83 | 46.89 | 55.21 | 36.77 |
| ✓ | | 52.08 | 51.12 | 57.03 | 37.10 |
| | ✓ | 50.02 | 48.49 | 56.51 | 36.99 |
| ✓ | ✓ | **53.55** | **51.19** | **58.37** | **37.28** |

object and binds accurate attributes such as color. By incorporating both $\mathcal{L}_{DSA}$ and $\mathcal{L}_{DSA}$, text-image alignment can be further enhanced.

**Effect of different level of semantic alignment.** DSAlign utilizes both local-level and global-level semantic alignment. To assess the effectiveness of semantic alignment at different levels, we conducted a detailed analysis using T2I-Compbench. Comparing the 2nd and 3rd rows of Table 6, it's evident that local-level

semantic alignment plays a more significant role in enhancing text-image alignment. This can be attributed to the fact that local-level semantic alignment provides region-wise alignment, enabling fine-grained supervision for the denoising process. As a result, the model can learn region information in a more detailed manner. Furthermore, by incorporating both local-level and global-level semantic alignment, we consistently achieve improvements in text-image alignment, which verifies the effectiveness of global-level semantic alignment.

**Table 7: Effect of different cross-attention layers.**

| Layer | Resolution | Color | Shape | Texture | Spatial |
|---|---|---|---|---|---|
| middle block | $8 \times 8$ | 49.50 | 50.02 | 55.48 | 36.24 |
| decoder layer | $64 \times 64$ | 51.60 | 49.85 | 56.58 | 35.21 |
| decoder layer | $32 \times 32$ | 49.76 | 49.10 | 55.95 | 35.79 |
| decoder layer | $16 \times 16$ | **53.55** | **51.19** | **58.37** | **37.28** |

**Table 8: Comparison to the state-of-the-art text-to-layout methods on NSR-1K.**

| | Numerical Reasoning | | | SpatialReasoning |
|---|---|---|---|---|
| Method | Precision | Recall | Accuracy | Accuracy |
| LayoutTransformer [17] | 75.70 | 61.69 | 22.26 | 6.36 |
| llama2-7B [47] | 75.42 | 90.47 | 71.00 | 30.51 |
| llama2-13B [47] | 76.90 | 92.80 | 77.56 | 29.15 |
| GPT-3.5 [2] | 94.81 | 96.49 | 86.33 | 86.33 |
| GPT-4 [2] | 88.23 | 97.60 | 94.48 | 90.11 |

**Effect of different cross-attention layers.** We analyze the effectiveness of employing different cross-attention layers with MAAlign as shown in Table 7. It's clear that MAAlign is most effective when applied to the $16 \times 16$ attention map obtained from the U-Net decoder, which is consistent with findings from previous works [18, 25].

### 4.7 Text-to-Layout Model Evaluation

To evaluate the performance of text-to-layout models, we examine the transformer-based model LayoutTransformer [17], as well as the latest large language models (LLMs) including GPT-4 [2], GPT-3.5 [2], Llama 2-7B [47], and Llama 2-13B [47]. We assess their ability to comprehend visual concepts through spatial and counting scenes in the NSR-1K benchmark. From table 8, we know that the GPT-4 outperforms all other models in both spatial and counting scenes, thus it is employed for layout planning in this work. Moreover, we observe that LLMs exhibit significantly superior performance compared to LayoutTransformer, highlighting its robust cross-modal spatial reasoning abilities. Comparing Table 2 and Table 8, we observe that the text-to-layout process is proficient, and the bottleneck of image generation in complex scenes primarily lies in layout-guided image control.

### 4.8 User Study

While quantitative metrics have limitations in providing a comprehensive assessment, we complement our analysis with a user study. We selected attributes including color, description, counting, position, and conflicts from Drawbench [42], resulting in 94 prompts. For each prompt, two images were generated by different models, which were then assigned to 20 individuals for evaluation. Evaluation of the images by the participants focused on two aspects: semantic alignment and aesthetic quality. Semantic alignment is utilized to assess whether the model can generate images that faithfully capture the semantic details from the input text prompts. Aesthetic quality is employed to determine if the images generated by the model exhibit any incoherent parts or unnatural poses.

Our method is compared to several baseline models including Stable Diffusion v1.4 [40], GLIGEN [26], Attend-and-Excite [9], and

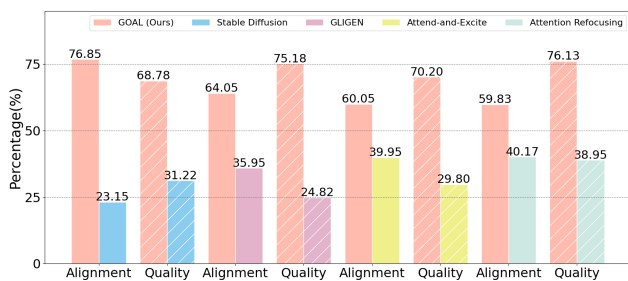

**Figure 6: User study on 94 prompts from Drawbench [42]. The ratios illustrate the participant preferences for the corresponding model. GOAL demonstrated superior performance in both image alignment and quality**

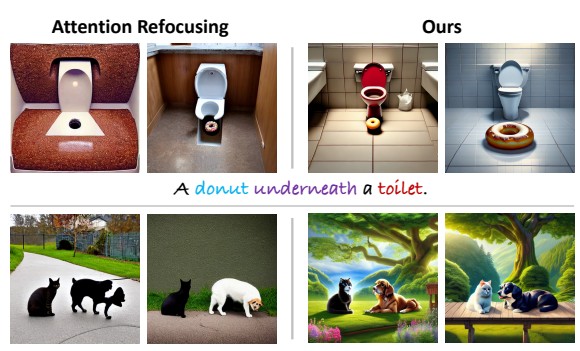

**Figure 7: Comparison with the training-free layout-aware method Attention Refocusing [34].**

Attention Refocusing [25]. Figure 6 shows that human preferences are consistent with our evaluation outcomes both in text-image alignment (*e.g.*, Table 1) and aesthetic quality (*e.g.*, Table 4), with 28.1% and 50.36% improvements compared to GLIGEN in alignment and quality respectively. Additionally, Figure 7 demonstrates that some images generated by Attention Refocusing [25] exhibit incoherent parts and unnatural poses, possibly attributed to the direct intervention of latent noise maps during sampling. In contrast, the proposed method enhances alignment without compromising image quality.

### 5 CONCLUSION

In this work, we propose Grounded jOint lAyout aLignment (GOAL) framework to handle Text-Image Misalignment issues in complex scenes. Discriminative semantic alignment (DSAlign) and masked attention alignment (MAAlign) are performed to provide explicit supervision with the layout conditions. In addition, We build a dataset GOAL2K to study the effectiveness of our alignment-based objectives, which composes 2000 semantically accurate image-text pairs and their layout annotations for model fine-tuning. Extensive experiments demonstrate that the proposed method achieves state-of-the-art performance on different benchmarks with improved image quality.

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
