# OpenReview forum: "GOAL: Grounded text-to-image Synthesis with Joint Layout Alignment Tuning"
_acmmm.org/ACMMM/2024/Conference — MM2024 Poster_

### Official Review · Reviewer_ir9q · 2024-05-05

**Rating:** 5
**Confidence:** 3

**Summary:**

This paper focuses layout control of text-to-image diffusion models with bounding-box conditions. It proposes a novel pipeline (GOAL) to improve the text-to-image alignment, as well as a dataset (GOAL2k) to evaluate their results. The author conduct extensive experiments on various benchmarks and demonstrated state-of-the-art performance.

**Strengths:**

Overall, this paper is well-written with clear presentation and figures. The motivation of the paper is clear: controlling the layout in diffusion models is one of the key challenges. Using a discriminative model to aid the region-wise alignment of text-to-image (T2I) models is a reasonable idea, but there is still room for improvement in terms of presentation. The experiments in this paper are comprehensive and well-support their claims. In particular, the authors demonstrate state-of-the-art quantitative results on three benchmarks, and the qualitative results appear good.

**Limitations:**

From my perspective, there are no severe issues in this paper, but there are some paragraphs that may appear unclear and can be improved.

1. In Sec 3.2 (L369-L377), the author states that overlapping bounding boxes are a challenge for previous layout control methods and proposes DSAlign to solve this issue. However, the author does not clearly present what "dissonant visual composition" is and whether this is an existing issue with references or a newly found problem with a clear definition and experiments for support. Moreover, the author did not show that their DSAlign can avoid this issue. It is true that the author showed DSAlign can ensure better region-wise semantic alignment in Tables 1, 2, and 3, but this does not necessarily mean that it avoids the specific issue caused by overlapping bounding boxes. I expect the author to provide experiments to support this claim.

2. In Sec 3.3, the author should define what "target region" and "irrelevant areas" are; otherwise, it appears unclear within its context.

3. In Sec 4.6 (L692-L694), the author states, "It can be attributed to the fact that discriminative loss provided by semantic alignment improves shape and texture metrics," which seems to lack linkage. The author should explain why the DSA loss can improve shape and texture or provide relevant references. Moreover, it seems that the $L_{pixel}$ and $L_{feature}$ in Table 5 are not defined.

4. This is not an issue, but the paper could benefit from discussing why spherical distance is the best choice, as it appears to deviate from the default cosine distance in the original CLIP.
---
Despite the above issues, the overall presentation and experiments in this paper are good. I would recommend a weak accept for this paper in its current form.

**Suitability:**

3

---

### Official Review · Reviewer_y8Kz · 2024-05-10

**Rating:** 4
**Confidence:** 4

**Summary:**

This paper proposes a novel framework named GOAL to handle Text-Image Misalignment issues in complex scenes. It improves the previous GLIGEN framework with two loss, $L_{𝐷𝑆𝐴}$ and $L_{𝑀𝑆𝐴}$. The proposed method optimize the model through text-image alignment in both the local masked part and the global attention map. Experimental results show the superior performance of GOAL.

**Strengths:**

1. The paper alleiviate the misalignment problem in attention map through local and global text-image alignment.
2. Both the qualitative and quatitative results show that the proposed method is more effective than other compared methods.
3. Ablation study also demonstrates the effectiveness the proposed modules.
4. The paper is well organized and easy to read.

**Limitations:**

1. In fig.5, the generating images are not so realistic, which might be due to the generation ability of the base model. The authors can improve the model performance using the other state-of-the-art models, such as sdxl.
2. In fig.5, the foreground objects and the background are not well integrated in the generating results.
3. Using $L_{𝐷𝑆𝐴}$ and $L_{𝑀𝑆𝐴}$, is the model able to generate customized concept? For example, can the model generate a specific object at the specific location given an object image?

**Suitability:**

3

---

### Official Review · Reviewer_isZJ · 2024-05-21

**Rating:** 3
**Confidence:** 2

**Summary:**

This paper proposes a framework for text-guided layout-to-image generation to address misalignment when generating complex scenes. To that end, a dataset called GOAL2k is constructed for fine-tuning. The model builds on GLIGEN and is extended through discriminative semantic alignment and masked attention losses. Extensive comparison with existing models visually and quantitatively demonstrates the approach's effectiveness.

**Strengths:**

- The paper is well-written and structured. Figures clearly visualize the approach and results.
- The proposed modules are well motivated and ablated.
- The results are convincing and many baselines are used for comparison.

**Limitations:**

Major:
- It looks like there is missing discussion and comparison with some of the mentioned related work [21], [42], [27], [53]
- Masked latents / cross-attention maps are already proposed in [27]; how are they similar/different to MAAlign in this paper?
- What is the intuition/motivation for the spherical distance that works best?
- Dalle-3 is used to generate the training data. How well does it perform when used for comparison (upper bound)?
- Are the comparisons against off-the-shelf baselines or also fine-tuned on the GOAL dataset? If they are not fine-tuned on the dataset, it is unclear whether the performance improvement over the baselines stems from the dataset or the modules.
- BoxDiff, a training-free method, compares very competitively without requiring any fine-tuning. How would it perform with fine-tuning?

Minor:
- Is there a typo between Table 3 and Table 4 on Attention Refocusing [34 vs 25]
- What are L𝑝𝑖𝑥𝑒𝑙 + L𝑓𝑒𝑎𝑡𝑢𝑟𝑒 losses? It seems that they were not introduced in the main text.

Justification for rating:
There are a few reasons for my rating despite the otherwise good impression. I am willing to improve my score upon clarification of the following:
1) The tables are not consistent. For example, some baselines appear in one table but not the other.
2) Some mentioned related work is not used as a baseline; see point above.
3) Is the masked attention alignment significantly new from the one in [27]?
4) Effect of fine-tuning: do baselines get a fair chance by also fine-tuning on the new dataset?

**Suitability:**

3

---

### Official Review · Reviewer_dwEx · 2024-05-24

**Rating:** 3
**Confidence:** 4

**Summary:**

The paper introduces the Grounded jOint lAyout aLignment (GOAL) framework, designed to address the Text-Image Misalignment problem in complex scene generation. GOAL incorporates two innovative modules: DSAlign focuses on low-level semantic alignment through discriminative tasks at the regional level, while MAAlign directs the model's attention towards the target object in the high-level feature space. They also created a dataset called GOAL2K to facilitate model fine-tuning. Extensive evaluations on various benchmarks. Experiments show substantial improvements over existing methods in color, shape, and texture metrics, highlighting GOAL's effectiveness in generating high-fidelity and semantically accurate images.

**Strengths:**

1. This paper designs two types of loss, which guide the generation of image layouts from the low level semantic and high level feature space semantic levels, and the calculation process do not rely on additional models. The framework is plug-and-play and is suitable for most text-to-image diffusion models.
2. The comparative experiment demonstrated in detail the advantages of this method compared to the previous training-free method, and showed the improvement in effectiveness after being applied to the layout-to-image method. Additionally, the ablation experiment is sufficient, involving loss combinations at different levels.

**Limitations:**

1. The paper adopts the CLIP model to adjust the low level semantics, which is based on single step denoising. However, the difficulty of denoising under different initial noise and text guidance varies. Therefore, the information of early single step denoising may be relatively blurry (being out of distribution for CLIP models) or even semantically unclear, and the loss calculated based on this may produce unstable effects.
2. DSAlign and MAlign loss based on feature space design are similar to previous methods, such as Training Free Layout Control with Cross Attention Guidance and Grounded Text to Image Synthesis with Attention Refocusing. Their core is to control the corresponding areas of graphics and text within the designated areas of the layout, which cannot be considered as a very innovative contribution.

**Suitability:**

3

---

### Meta-Review · Area_Chair_1dvd · 2024-07-02

**Recommendation:** Accept (Poster)
**Confidence:** 4

**Metareview:**

The initial ratings are 2 BR, 1 BA, and 1 WA. After rebuttal, they are 1 BR, 2 BA, and 1 WA. Most reviewers appreciated the proposed method about text-guided layout-to-image generation. One reviewer is still not satisfied with the authors' responses for the explanation to the loss. The concern, however, is minor compared to the contribution of the paper. The decision is, therefore, to accept the work. The authors are encouraged to incorporate the material from their response to the camera ready version.